# Quality of Care: Ecological Study for the Evaluation of Completeness and Accuracy in Nursing Assessment

**DOI:** 10.3390/ijerph17093259

**Published:** 2020-05-07

**Authors:** Angela Iula, Carola Ialungo, Chiara de Waure, Matteo Raponi, Matteo Burgazzoli, Maurizio Zega, Caterina Galletti, Gianfranco Damiani

**Affiliations:** 1Degree in Nursing & Midwifery Sciences, Università Cattolica del Sacro Cuore, Largo Francesco Vito, 1, 00168 Rome, Italy; caro.ialungo@gmail.com (C.I.); caterina.galletti@unicatt.it (C.G.); 2Department of Experimental Medicine, University of Perugia, 06123 Perugia, Italy; chiara.dewaure@unipg.it; 3Internal Audit, Fondazione Policlinico Universitario A. Gemelli IRCCS, 00168 Rome, Italy; matteo.raponi@guest.policlinicogemelli.it; 4Internal Control System and Auditing Functions, Università Cattolica del Sacro Cuore, 20123 Milan, Italy; matteo.burgazzoli@unicatt.it; 5Department UOC SITRA, Fondazione Policlinico Universitario A. Gemelli IRCCS, 00168 Rome, Italy; maurizio.zega@policlinicogemelli.it; 6Department of Woman and Child Health and Public Health, Fondazione Policlinico Universitario A. Gemelli IRCCS, 00168 Rome, Italy; gianfranco.damiani@unicatt.it; 7Department of Health Sciences and Public Health, Università Cattolica del Sacro Cuore, 00168 Rome, Italy

**Keywords:** nursing records, nursing assessment, pain measurement, nutrition assessment, quality of healthcare

## Abstract

Nursing documentation is an important proxy of the quality of care, and quality indicators in nursing assessment can be used to assess and improve the quality of care in health care institutions. The study aims to evaluate the completeness and the accuracy of nursing assessment, analyzing the compilation of pain assessment and nutritional status (body mass index (BMI)) in computerized nursing records, and how it is influenced by four variables: nurse to patient ratio, diagnosis related group weight (DRG), seniority of charge nurse, and type of ward (medical, surgical or other). The observational ecological pilot study was conducted between September and October 2018 in an Italian Tertiary-Level Teaching Hospital. The nursing documentation analyzed for the ‘Assessment’ phase included 12,513 records, 50.4% concerning pain assessment, and 45% BMI. The nurse–patient ratio showed a significant direct association with the assessment of nutritional status (*p* = 0.032). The average weight DRG has a negative influence on pain and BMI assessment; the surgical units positively correlate with the compilation of nursing assessment (BMI and pain). The nursing process is an essential component for the continuous improvement in the quality of care. Nurses need to be accountable to improve their knowledge and skills in nursing documentation.

## 1. Introduction

The problem of measurement of nursing care and the related health outcomes is widely debated in the literature [1] and is of particular interest in the nursing research [2]. Quality of care is frequently measured by using quality indicators. Nursing documentation provides an important proxy of the quality of care for hospitalized patients [3] and is a significant indicator of effective patient care delivery.

Nurses have considerable interest in increasing the quality of nursing documentation [4]. It is important to provide the patient’s conditions to allow the continuity of the care and improve patient outcomes [5]. To achieve these purposes, nursing documentation needs to hold valid and reliable information and comply with established standards [6].

The Joint Commission International [7] promotes the creation of a culture of safety and quality in the institutions to continually improve patient care processes. In this context, it is worth pointing out that digital health records are relevant information sources to monitor the achieved quality and safety levels. Therefore, in the nursing record system, the documentation is considered essential to generate benefits for individualized care planning and for the evaluation of care in general [8].

According to Wang, Yu, and Hailey [9], the quality of nursing documentation includes three main components: content, documentation process, and format or structure. Documentation content focuses on completeness and accuracy of data that reflects reality. The documentation process focuses on the patient’s data completeness and the regularity of data in the patient’s records. The documentation’s structure focuses on physical presentation, which includes the legibility and completeness of the patient’s information.

Quality of care is strongly linked to the documentation of nurses’ work, and the nursing process can be monitored by evaluating some quality indicators in nursing documentation related to the accuracy of initial assessment, nursing diagnosis, and nursing interventions.

The registered nurses (RN) are responsible and accountable for the nursing documentation that is used throughout an organization but often appears incomplete, especially in the compilation of certain areas, such as assessment, nursing diagnoses, and nursing interventions [8]. In particular, the present study will consider, in nursing assessment, two important entries for the patient: pain assessment and nutritional status (BMI, body mass index) that nurses must complete for the patient.

The Joint Commission International defines, in its quality standards, that “all inpatients and outpatients are screened for pain and assessed when pain is present” (AOP 1.5) [7]. In Italy, a law provides citizens with the right to be assessed and access pain therapy [10]. The health care systems have invested resources to achieve these goals. Indeed, in all clinical documentation (medical and/or nursing) used in health facilities, assessed pain, the characteristics of the detected pain and its evolution during the hospitalization, as well as the technique antalgic and the drugs used, the relative dosages and the antalgic result achieved, must be reported [10].

Standard-setting organizations use pain management documentation as a key indicator of quality of care [11]. Nurses play a crucial role in pain assessment and management. They often act as mediators between the doctor and the patient and serve as the main observer of pain and discomfort in the patient.

National and international organizations [12] recommend nutritional risk screening for all the patients admitted to hospital, to identify who would likely benefit from relevant, individualized nutritional care [13].

BMI is a useful and reliable measure of the appropriateness of weight related to height, which is simple to carry out and is well correlated with body-fat percentage. All patients should have the height and weight information recorded as part of their nursing or medical assessment on admission to the hospital [14]. Nurses have close and continuous contact with patients, so they are in an ideal position to screen for malnutrition [15] and to document these results in a nursing record system.

The objective of this study was to evaluate the completeness and the accuracy of nursing assessment (e.g., pain assessment and nutritional status) in computerized nursing records, and how it is influenced by four variables identified in the literature and associated to the quality of nursing care. These variables were nurse to patient ratio (nursing care hours per patient day, HPPD) [2]; diagnosis-related group weight (DRGs has a weight which reflect the average service intensity for the patient with that illness) [16]; seniority of the charge nurse (length of role as measured by date of hire) [17]; type of ward (medical unit, surgical unit, or other unit providing specialty care) [2,18].

## 2. Materials and Methods

The study was an observational ecological pilot study conducted in an Italian tertiary level teaching hospital, whose main characteristics are size (≥1000 hospital beds), teaching status, high technology medical equipment. This type of epidemiological study allowed us to use the ward as a single sample unit since all patients hospitalized in one of them were exposed to the same variables. This study was conducted in two months, from September to October 2018.

The analysis was performed using the computerized nursing records, located in the management information system, to support nurses in clinical reasoning. Indeed, nursing staff enter the patient’s data directly into the computer and can refer to nursing diagnosis and standard care plan interventions. The electronic records system was implemented by this teaching hospital to monitor and manage patient care and to file an indication regarding patient stay in the wards.

We have taken into consideration the nursing documentation of all patients admitted to the teaching hospital’s wards over a twelve-month period, between 2016−2017. The following inclusion criteria were considered: wards where computerized nursing records were used, ordinary hospitalization of SSN (Sistema Sanitario Nazionale) (excluded week surgery), and length of stay ≥3 days.

The pain and BMI items of the nursing assessment were extracted by a computerized nursing system from a data-warehouse and associated with an identification number referable to the ward but not to the patient. The data were included in a database and related to four variables (Table 1). 

The DRG average weight was extracted through the information technology (IT) system and was referred to admissions during the two-year period, 2016−2017.

The nurse–patient ratio was calculated through the evaluation of the number of daily nurses’ hours/number of patients (HPPD).

The seniority of the charge nurse was evaluated by calculating the years of seniority as charge nurse starting from the year in which he/she took the charge role. For this reason, four categories were defined (Table 2): seniority equal to or greater than 20 years, seniority from 19 to 12 years, seniority from 11 to 4 years, seniority equal or less than 3 years.

The wards, from which the electronic nursing documentation samples to be analyzed were extracted, were divided into three categories: medical units, surgical units, other units providing specialty care (medical and surgical units).

Aggregated data were analyzed anonymously, according to the privacy protection criteria already in force at the hospital.

### 2.1. Statistical Analysis

The descriptive statistical analysis was conducted using absolute and relative frequencies for pain and BMI parameters for each unit.

A linear regression model was run to assess the role of each variable in predicting the percentage of cases with the assessment of pain and BMI level done. Model goodness of fit was assessed through R^2^, and outliers were evaluated based on Cook’s distance. Collinearity was assessed by the variance inflation factor. Results were reported as beta coefficients with 95% confidence intervals (95% CI). All statistical analysis was performed using SPSS (IBM Corp, Armonk, NY, USA) v. 22.0.

### 2.2. Ethics Approval and Consent to Participate

This study was approved by the IRB of our institution, July 19th, 2018, Prot. 28982/18 and Prot. 40229/18 ID: 2174.

## 3. Results

Nursing documentation analyzed for the assessment phase was 12,513 records, with a mean of 417 per ward. The indicator pain assessment was found in 6307 (50.4%) records of the total number of nursing records analyzed. The indicator nutritional status (BMI) was found in 5631 (45%) records of the total number of nursing records analyzed.

### 3.1. Results of the Four Variables

#### 3.1.1. Type of Ward

We analyzed 30 units of teaching hospital, divided as follows: 15 medical units (50%), 7 surgical units (23%), 8 other units providing specialty care (27%).

#### 3.1.2. Seniority of Nurse Coordinator

We analyzed the seniority of charge nurse in 30 units, and we obtained the following information: 7 charge nurses with seniority equal to or greater than 20 years (23%); 5 charge nurses with seniority from 19 to 12 years (17%); 13 charge nurses with seniority from 11 to 4 years (43%); 5 charge nurses with seniority equal to or less than 3 years (17%).

#### 3.1.3. DRG Mean Weight

The DRG mean weight ranged from a minimum of 0.72 to a maximum of 3.18, with a mean weight of 1.43 and a standard deviation of 0.49.

#### 3.1.4. Nursing–Patient Ratio

The nursing care hours per patient per day ranged from a minimum of 1.57 to a maximum of 4.8, with mean weight of 2.91 and a standard deviation of 0.74.

### 3.2. Results of Linear Regression Model

With respect to the first endpoint, i.e., the percentage of cases with the assessment of pain (Table 3), none of the independent variables showed an association. No collinearity was detected among independent variables, but four potential outliers were identified, namely Hematology, Pediatric Neuropsychiatry, Neurology, Pediatric Day Hospital. Nonetheless, the removal of these two observations did not change the main results. As far as the cases with the assessment BMI were concerned, the nurse–patient ratio showed a significant association (Table 4). No collinearity was detected, but the same two potential outliers were also identified, but results were confirmed after their removal.

## 4. Discussion

The information received from the nursing documentation can be used as a proxy of the care given [20] and of the nursing process [21].

The need to monitor nursing care quality, which is very important for the management of a tertiary level teaching hospital, may be driven by systematic data collection. The innovation and adoption of digital technologies allowed us, in our study, to select and report indicators to demonstrate the quality of care delivered.

The availability of computerized nursing records allowed us the important opportunity to access a large amount of data (12,513 records). The results of our study show that pain assessment and BMI have a low frequency of compilation (50.4% for pain evaluation and 45% for BMI).

This result is consistent with other studies reporting a lack of nursing data in the patient’s health care documentation [22,23]. In addition, Shugarman et al. [24] found that half of all nursing staff used informal screening approaches rather than the numeric rating scale. Khalatbari-Soltani and MarquesVidal [25] showed poor adherence to guidelines regarding nutrition monitoring and reporting. It could be due to the patient’s characteristics and to the lack of training, staff, and time or to the impossibility to find these activities by extracting the data from the data warehouse. In contrast, in the literature, the computerized nursing record is a useful tool to speed up the speed evaluation processes when complete and accurate documentation is available [26]. However, these results are consistent with studies showing that multiple factors can affect the quality of nursing documentation (e.g., nurses’ levels of education and training; the amount of daily care provided by registered nurses; nurse characteristics, such as years of work, age, or greater contact with patients) [21,24].

The data collected for our study did not allow us to evaluate in depth the motivations that led such a result; a hypothesis, that should be tested in a subsequent research, could be an incorrect use of the informatics tool (patient pain report and/or patient nutrition report) in the nursing assessment phase.

Significant results emerged with respect to the association between the nursing ratio and the compilation of the BMI status. Increasing the hours of nursing per patient day significantly increased the correct compilation of BMI in nursing documentation. This finding is related to the nurse being given the opportunity to dedicate enough time to the patient’s clinical assessment phase. Indeed, the main barriers to an efficient screening of nutritional status were nurses’ lack of time and poor knowledge of nutrition care processes [25].

The other parameter analyzed (assessment of pain) did not show any significant associations between the other variables of our study. Literature showed there are some organization-related barriers that prevent nurses from performing pain assessment and management, particularly high patient-to-nurse ratio and lack of psychosocial support services [24,27].

However, a negative correlation was found between pain assessment and BMI status with DRG average weight. This means that higher DRG values were linked to lower missed compilation of the two parameters. It is possible that patients, who were more acutely ill, got more overall attention than those who were less acute. It is difficult to compare this finding across different studies. Indeed, the studies do not consider the DRG, but the case mix index (ICM) [28]; it may represent the relative differences in resource expended for patient care. A recommendation for future research could be to enrich the computerized folders with these data to compare the different performances of the two variables.

To assess the quality and safety of nursing care, it is necessary to structure a system that can measure it by means of a set of indicators sensitive to nursing care [29]. Nursing documentation plays an important role in the delivery of quality nursing care services, and the nursing process helps to regularly monitor and to improve their quality [30].

The evaluation of completeness in nurses’ assessment has been the subject of numerous studies, including the Missed Nursing Care (MNC) [31]. MNC is defined as any aspect of required patient care that is omitted (in part or whole) or delayed [32].

The choice to complete, delay, or omit items of the nursing process is influenced by four factors internal to a nurse: team norms, decision-making processes, internal values and beliefs, and habits [32]. The main reasons for missing this care are labor resources, material resources, and communication/teamwork, which interact with the nursing process, and are filtered by the nurses’ internal process [31].

There is good evidence that MNC influences patient outcomes negatively. For example, Schubert et al. [33] showed that higher missed nursing care was linked to increased patient pain and discomfort. Green et al. [34] also revealed that nurses rarely use screening tools for identifying patients at risk of malnutrition. Indeed, nurses may perceive professional judgment to be a useful or more accurate screening tool. On the subject of professional judgment, Schubert et al. [33] showed that implicit rationing was associated with a significant threat to patient safety and quality of care.

Therefore, further studies may be useful to investigate the impact of rationing of nursing care on completeness and accuracy of nursing documentation.

### Study Limitations

There are some potential limitations to our study. 

Data were collected in a single teaching hospital, and concern relating to generalizability and comparability of the results may arise.

The choice to consider the hours of nursing care, without calculating the differences between the units with nursing assistants, whose presence changes in a significant way the skill mix of the nursing team may limit the study.

The seniority of charge nurse can be considered a useful variable to analyze, but it could be integrated with the evaluation of the education of the whole nursing team.

This type of study does not allow the evaluation of the cause–effect connection.

## 5. Conclusions

The findings of this study explain, at least in part, what is occurring within the process of proving nursing care. It revealed the fact that aspects of nursing care are not being completed. The findings of our study indicate that the compilation of pain assessment and BMI status on nursing documentation is low. Our results substantiated other investigations that have shown that specific aspects of care are being missed. This could be associated with excessive workload, inadequate knowledge, or simply trust in their own clinical judgment rather than a tool to asses real risk.

The quality and safety of nursing care are based on a system that can measure a series of indicators sensitive to nursing care. Nurse-ratio had a statistical significance (*p* = 0.032) with the compilation of nutritional status (BMI). This was correlated to the possibility of nurses to dedicate more time to the assessment of patients. Nurse-ratio also had a positive correlation with the other variables analyzed, making it an interesting theme for managerial evaluation in the healthcare system. The nursing care planning process is an essential component to be analyzed from a perspective of continuous improvement in the quality.

It is important to implement the use of computerized nursing documentation since it helps to validate what nurses do, how they do it, and the impact that this has on their patients and the quality of care delivered.

## Figures and Tables

**Table 1 ijerph-17-03259-t001:** Description of study variables.

Variables	Description
DRG average Weight	Admissions from October 2016 to November 2017
Nurse–patient Ratio	HPPD = Nurses/Patients = Productive Nurses’ Hours/Patient Days × 24
Seniority of Charge Nurse	Years of Seniority as a Charge Nurse
Type of Ward	1. Medical Units 2. Surgical Units 3. Other Units Providing Specialty Care (Medical and Surgical Units)

**Table 2 ijerph-17-03259-t002:** Rationale of the seniority of the charge nurse

Categories	Seniority ≥ 20 Years	Seniority from 19 to 12 Years	Seniority from 11 to 4 Years	Seniority ≤ 3 Years
Seniority	Charge nurse who have been in charge role before 1996.	Charge nurse who has been in charge role between the years 1997 and 2004.	Charge nurse who has been in charge role between the years 2005 and 2012.	Charge nurse who has been in charge role between the years 2013 and 2016.
Characteristic Elements *	Tendency to challenge authority. Open and direct communication, group meetings, recognition of the work done.	Professional autonomy, social recognition of the profession, and welfare models. Interacts with the institutions.	Professional culture oriented to the personalization of Assistance and health results. They prefer intelligent leadership and effective mentors.	Global generation for the use of technology, tends to group activities and to immediate communication and appreciates supervision.

* Adapted [19].

**Table 3 ijerph-17-03259-t003:** Percentage of cases with the assessment of pain.

Variable	B Coefficient	*p*-Value	95% CI
Seniority from 19 to 12 yr vs. Seniority ≥ 20 yr	0.168	0.400	−0.238; 0.574
Seniority from 11 to 4 yr vs. Seniority ≥ 20 yr	0.050	0.730	−0.246; 0.346
Seniority ≤ 3 yr vs. Seniority ≥ 20 yr	0.081	0.645	−0.277; 0.438
DRG Mean Weight	−0.070	0.618	−0.355; 0.216
Nurse–patient Ratio	0.096	0.266	−0.078; 0.269
Surgical vs. Medical Units	−0.091	0.540	−0.213; 0.395
Other Units vs. Medical Units	−0.053	0.714	−0.346; 0.241
R^2^ = 0.127			

R^2^ is the proportion of the variance in the dependent variable that the independent variables explain collectively. It measures the strength of the relationship between the regression model and the dependent variable on a convenient 0–1 scale.

**Table 4 ijerph-17-03259-t004:** Percentage of cases with the assessment BMI.

Variable	B Coefficient	*p*-Value	95% CI
Seniority from 19 to 12 yr vs. Seniority ≥ 20 yr	0.139	0.544	−0.329; 0.680
Seniority from 11 to 4 yr vs. Seniority ≥ 20 yr	0.076	0.648	−0.265; 0.417
Seniority ≤ 3 yr vs. Seniority ≥ 20 yr	0.032	0.872	−0.380; 0.445
DRG Mean Weight	−0.239	0.146	−0.568; 0.090
Nurse–patient Ratio	0.220	0.032	0.020; 0,420
Surgical vs. Medical Units	0.307	0.083	−0,043; 0.657
Other Units vs. Medical Units	0.107	0.519	−0.231; 0.445
R^2^ = 0.298

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
