# Peer review of "Quality of Care: Ecological Study for the Evaluation of Completeness and Accuracy in Nursing Assessment"

_ijerph, 2020, doi:10.3390/ijerph17093259_

Round 1

Reviewer 1 Report

Dear authors, 

Thank you for an interesting and thoughtful manuscript. I suggest some minor comments:

  1. I don't understand the concept of "medical units"- all of the units at hospitals relate to medical treatment. 
  2. Please reword one of the sentences in the abstract (you have two): Nursing documentation is an important proxy of the....
  3. In the tables, Please justify the column of variables to left (don't center the words)
  4.  In the tables, It is acceptable to indicate 95%CI with a dash between the numbers. (e.g. -0,380 - 0,445). Please check the journal policy before changing. 
  5. When indicating the total number of records, please use a comma instead of a period.  12,513

All my best,

Author Response

Dear Reviewer,

Thank you for your comments.

Point 1: I don't understand the concept of "medical units"- all of the units at hospitals relate to medical treatment. 

Response 1: In our study, we differentiate medical units, that provide non-intensive care to adult patients with a variety of general medical needs, from surgical units, that provide intensive care for critically ill patients.

Point 2: Please reword one of the sentences in the abstract (you have two): Nursing documentation is an important proxy of the...

Response 2: Thank you for your comment, we reword the sentences in the abstract (row 38-39).

Point 3: In the tables, Please justify the column of variables to left (don't center the words)

Response 3: We justify the columns of variable to left in all the tables (Table 1, Table 2, Table 3 and Table 4).

Point 4: In the tables, It is acceptable to indicate 95%CI with a dash between the numbers. (e.g. -0,380 - 0,445). Please check the journal policy before changing.

Response 4: Please note that ‘dash -’ could be confuse with ‘minus -’.

Point 5: When indicating the total number of records, please use a comma instead of a period. 12,513

Response 5: We made the change so we use a comma 12,513 (row 34, row 225).

Reviewer 2 Report

This paper is a study to evaluate the completeness and the accuracy of nursing assessment in computerized nursing records, and how it is influenced by four variables. Unfortunately, in this study is not clearly indicated.

1. The Introduction, Literature Review and Hypotheses is quite exhaustive. However, some concepts are repeated too many times, therefore making the readability of this part tricky. Please, check and delete some of them.

2. Methodology needs to be clearly specified such as how the research was conducted and why.
: The authors should present and highlight the original ideas and contributions in the manuscript.

3. The implications from the study may be impeded by specific methodological issues.

4. The scope and importance of this opinion is not clear enough.
: The opinion is too dense and could strongly benefit from a more elaborate explanation of terminology and concepts.

5. In my opinion, this represents the main issue of the manuscript. Indeed, there is complete absence of comparison with the existing literature, making this work completely standalone with respect to the field of investigation it belongs to, to the reader's eye. Please, provide some literature comparison, maybe with other traditional and innovative healthcare schemes

6. Overall, English language and grammar should be carefully revised by a native speaker and several typos are present throughout the text and should be corrected.

Author Response

Dear Reviewer,

Thanks for your comments.

Point 1: The Introduction, Literature Review and Hypotheses is quite exhaustive. However, some concepts are repeated too many times, therefore making the readability of this part tricky. Please, check and delete some of them

Response 1: We reviewed the introduction deleting some repeated concept (row 61-62, row 69-71).

Point 2: Methodology needs to be clearly specified such as how the research was conducted and why. : The authors should present and highlight the original ideas and contributions in the manuscript.

Response 2: We revised all section “Material and Methods” to clarify the methodology we used to conduct the study. We highlighted the original ideas and contributions in the manuscript in row 220-225 and 316-318.

Point 3: The implications from the study may be impeded by specific methodological issues.

Response 3: We revised all section “Material and Methods” to clarify the methodology we used to conduct the study. We highlighted the original ideas and contributions in the manuscript in row 220-223 and 316-318.

Point 4: The scope and importance of this opinion is not clear enough.

The opinion is too dense and could strongly benefit from a more elaborate explanation of terminology and concepts

Response 4: We apologize, we tried to better explain terminology and concepts

Point 5: In my opinion, this represents the main issue of the manuscript. Indeed, there is complete absence of comparison with the existing literature, making this work completely standalone with respect to the field of investigation it belongs to, to the reader's eye. Please, provide some literature comparison, maybe with other traditional and innovative healthcare schemes

Response 5: We specified the bibliographic references associated with each variable identified (row 102-105).

We reviewed discussion (row 220-225; 242-245; 268-271).

Point 6: Overall, English language and grammar should be carefully revised by a native speaker and several typos are present throughout the text and should be corrected

Response 6: A native speaker revised the paper for the English language. In fact, there are different correction in the article.

Reviewer 3 Report

Dear Autors,   I have read the article on nursing documentation with great interest. This is an extremely important issue in terms of the quality of nursing care. Authors should consider completing the content with the following elements:   Introduction Providing information on whether all hospitals in Italy have computerized nursing records. And possibly, if similar tests were carried out in hospitals that do not use computer systems.   Reference and method Authors should present more information on other types of hospitals in the healthcare system in Italy. And, if necessary, indicate the differences between individual reference degrees, especially in the context of nursing documentation. By definition, hospitals of the third degree of reference (university, clinical) should pay more attention to the completeness of documentation, including the nursing documentation.   Discussion Please indicate the limitations of the research.   Conclusions Please do not refer to the results of other studies in the conclusions .

Author Response

Dear Reviewer,

Thanks for your comments.

Point 1: Introduction Providing information on whether all hospitals in Italy have computerized nursing records. And possibly, if similar tests were carried out in hospitals that do not use computer systems.

Response 1: We entered additional information (row 114-117; 220-223; 316-318).

Point 2: Reference and method Authors should present more information on other types of hospitals in the healthcare system in Italy. And, if necessary, indicate the differences between individual reference degrees, especially in the context of nursing documentation. By definition, hospitals of the third degree of reference (university, clinical) should pay more attention to the completeness of documentation, including the nursing documentation.

Response 2: We revised the article including information about Teaching Hospital in row 109-110; 74-76.

Point 3: Discussion Please indicate the limitations of the research.

Response 3: We indicated the limitations of the research at row 289-298.

Point 4: Conclusions Please do not refer to the results of other studies in the conclusions

Response 4: We deleted the result of the study in conclusion (row 308-309).

Round 2

Reviewer 2 Report

The paper you submitted has been revised well.

Please update your reference.

I recommend English proofreading.